# Applicable Life-History and Molecular Traits for Studying the Effects of Anhydrobiosis on Aging in Tardigrades

Amit Kumar Nagwani [1], Łukasz Kaczmarek [2] and Hanna Kmita [1,*,†]

1 Department of Bioenergetics, Faculty of Biology, Adam Mickiewicz University in Poznań, Uniwersytetu Poznańskiego 6, 61-614 Poznan, Poland

2 Department of Animal Taxonomy and Ecology, Faculty of Biology, Adam Mickiewicz University in Poznań, Uniwersytetu Poznańskiego 6, 61-614 Poznan, Poland

* Correspondence: hanna.kmita@amu.edu.pl; Tel.: +61-829-5902

† Current Address: Department of Bioenergetics, Institute of Molecular Biology and Biotechnology, Faculty of Biology, Adam Mickiewicz University in Poznań, Uniwersytetu Poznańskiego 6, 61-614 Poznan, Poland.

**Abstract:** Anhydrobiosis is induced by loss of water and indicates dehydration tolerance. Survival of dehydration is possible through changes at different levels of organism organization, including a remarkable reduction in metabolic activity at the cellular level. Thus, anhydrobiosis may be regarded as an anti-aging strategy. Accordingly, two hypotheses named after popular stories, "Sleeping Beauty" and "The Picture of Dorian Gray", were proposed to explain the effect of anhydrobiosis on aging. The two hypotheses predict the presence (The Picture of Dorian Gray) or absence (Sleeping Beauty) of observable aging symptoms for animals undergoing anhydrobiosis. Predictions of these hypotheses have rarely been tested, and the cellular level has not been addressed. Tardigrades appear to be a useful model for studying the effect of anhydrobiosis on aging, as they are able to enter and survive anhydrobiosis at any stage of life, although not with the same success for all species. In this review, we discuss anhydrobiosis and aging mechanisms as well as tardigrade diversity and indicate possible multilevel markers that can be used to study the impact of anhydrobiosis on tardigrade aging. This review provides data on tardigrade diversity that may also be useful for human aging studies.

**Keywords:** tardigrades; anhydrobiosis; aging; Sleeping Beauty; The Picture of Dorian Gray; life-history traits; cellular traits

## 1. Introduction

Tardigrades (water bears) are microinvertebrates found in marine, freshwater, and limno-terrestrial habitats [1]. The number of known tardigrade species has been steadily increasing over the past decades. Currently, 1019 freshwater and 217 marine species are reported in the World Register of Marine Species (WoRMS) database, and ca. 1400 species are described in the Actual Checklist of Tardigrada Species (41st edition: 16 May 2022) [2,3]. The phylum is divided into two classes, Eutardigrada and Heterotardigrada, distinguished mainly on the basis of claws, dorsal and cephalic cuticles, body appendages, and reproductive structures [4].

Tardigrades differ in reproduction modes; they can be dioecious, parthenogenetic, or hermaphroditic [5]. The known life-history traits of tardigrades, including total lifespan, number of molts, hatching time, and hatching success, do not appear to be strictly correlated with a certain reproduction mode and vary between species [6]. A total lifespan ranging from several weeks to several years has been observed in different tardigrade species with similar reproductive modes [7]. Food preferences are also diverse in tardigrades, including plant cells, algae, bacteria, nematodes, rotifers, or other tardigrades [8–10]. Tardigrades differ in their ability to survive in extreme environmental conditions through cryptobiosis [11]. Several types of cryptobiosis are distinguished according to the triggering factor: anhydrobiosis (lack of water), cryobiosis (low temperature), anoxybiosis (lack of

oxygen), and osmobiosis (high or low osmotic pressure) [12–15]. During cryptobiosis, tardigrades reduce their metabolic activity, restoring it when conditions again become favorable [16]. Thus, cryptobiosis may extend their lifespan by many years [17,18].

One of the most important impacts of cryptobiosis on tardigrade lifespan is its impact on aging. For anhydrobiosis, the prevalent form of cryptobiosis [19], two hypotheses, denoted as "Sleeping Beauty" and "The Picture of Dorian Gray", were proposed [20,21] to explain its effect on aging. The "Sleeping Beauty" hypothesis assumes complete exclusion of the time spent in anhydrobiosis; aging does not occur. The "The Picture of Dorian Gray" hypothesis predicts that the anhydrobiotic organism ages, at least in the initial stages of anhydrobiosis, such that aging proceeds or is slowed down [19]. Thus, the time spent in anhydrobiosis increases ("Sleeping Beauty") or does not increase ("The Picture of Dorian Gray") the lifespan of anhydrobiotic animals compared with non-dehydrated (active) animals, possibly due to differences in metabolic rate and protection against aging-imposed damages, although this has not been analyzed. Predictions of these hypotheses have rarely been tested. The lifespan of active specimens is currently the main parameter used to verify the hypotheses; total or age-specific fecundity, specimen vitality, and—rarely—morphology are also used [22]. The "Sleeping Beauty" hypothesis seems to apply to the bdelloid rotifers *Macrotrachela quadricornifera* Milne, 1886 [22] and *Adineta ricciae* Segers & Shiel, 2005 [20]; the free-living nematode *Panagrolaimus rigidus* Schneider, 1866 appears to follow "The Picture of Dorian Gray" [19]. For tardigrades, only one species (*Milnesium tardigradum* Doyere, 1840) [17] has been studied in this context; it was shown to follow the "Sleeping Beauty" hypothesis. The "Sleeping Beauty" hypothesis, in relation to the effect of anhydrobiosis on aging, seems to support complete suspension of metabolism [23]. However, respiration-based metabolism was detectable in anhydrobiotic animals at a low level for the tardigrade *Macrobiotus hufelandi* C.A.S. Schultze, 1834 [24] and the stem nematode *Ditylenchus dipsaci* (Kühn, 1857) [25,26]. Moreover, the activity of a mitochondrial protein known as alternative oxidase (AOX) during dehydration likely contributes to anhydrobiosis survival of *Milnesium inceptum* Morek, Suzuki, Schill, Georgiev, Yankova, Marley, & Michalczyk, 2019 [27]. Considering the available data, verification of these hypotheses remains an intriguing possibility. More research is required to determine the effect of anhydrobiosis on animal aging. However, markers that can verify these aging hypotheses are limited. The life-history traits could be used, but understanding the impact of anhydrobiosis on aging also requires study at the cellular level.

Extensive previous investigations including comparative genomics, transcriptomic analysis, and proteomic analysis have provided useful information concerning the mechanism of anhydrobiosis in several tardigrade species [16,28–30]. However, they indicated a high degree of divergence of these mechanisms among tardigrade species, suggesting unique molecular adaptations [29–31]. Previous studies focused on the identification of genes and encoded proteins involved in anhydrobiosis, and information related to tardigrade aging is minimal. Thus, we present available data on tardigrade life-history traits; highlight the features useful for studying the effect of anhydrobiosis on aging; and indicate the cellular traits that could serve as markers for analysis. Successful identification of markers will help to explain the anhydrobiosis effect in aging and contribute to a better understanding of cell death and the development of applicative solutions. Moreover, differences in anhydrobiosis ability observed for tardigrade species provide a great opportunity for research on the involved mechanisms.

## 2. Anhydrobiosis

Anhydrobiosis indicates "life without water" and is also known as "dehydration tolerance". Anhydrobiosis is induced by loss of water. As with other invertebrates, such as nematodes and rotifers, tardigrades exhibit a remarkable ability to enter and survive in an anhydrobiotic state at any stage of life [12,32]. The apparent decrease in metabolism with low water content is the most challenging aspect of anhydrobiosis. The relationship between hydration and metabolic rate, and whether anhydrobiotic animals

should be classified as living (metabolically active) or dead (ametabolic), has been debated. However, despite years of research on anhydrobiotic invertebrates (e.g., [23,31–33]), the metabolic status and preservation of molecular integrity with low water content are not completely understood.

During anhydrobiosis, tardigrades form a tun-shaped structure to reduce their evaporation surface [16]. This ability is present in all tardigrade lineages, including marine echiniscoideans and arthrotardigrades, indicating that it is an ancient and homologous trait and a morphological and behavioral adaption to dehydration [16,34,35]. The process of tun formation is generally accompanied by contraction of the longitudinal intersegmental cuticle and invagination of the legs [36,37]. Intracellular lipids may be responsible for reduced transpiration rates and decreased cuticle permeability [38]. Tun formation is an active process requiring energy supply; thus, only active animals with functional mitochondria can achieve it [16,38]. Species inhabiting different microenvironments often exhibit differences in tun formation. For instance, limno-terrestrial species usually form tuns within half an hour; marine-tidal species may accomplish it in seconds [39]. A study of *Echiniscoides sigismundi* (M. Schultze, 1865), a marine tardigrade species, revealed that tun formation is not a prerequisite for dehydration tolerance in all tardigrade species and may be an adaptation to elevated external pressure rather than desiccation [35]. Moreover, it was reported that marine and true freshwater tardigrades cannot survive dehydration and undergo anhydrobiosis [7].

Dehydration generally causes severe damage to cellular structures, resulting in cell death; tardigrades have the ability to withstand such extremes. Available data indicate that tardigrade resistance to dehydration is based on mechanisms highly conserved within eukaryotes and mechanisms specific to the animals [28,30]. These mechanisms are mediated by oxidative stress response proteins (superoxide dismutase glutathione peroxidase, glutathione reductase, glutathione transferase, and catalase), chaperones (heat shock proteins), DNA repair enzymes (recombinases involved in DNA homologous recombination), water transporters (aquaporins), and intrinsically disordered proteins, such as late embryogenesis abundant proteins (LEA) and tardigrade-specific proteins, including tardigrade-specific intrinsically disordered proteins (TDP) and damage suppressor proteins (Dsup) [13,30,40,41]. Available data indicate that the mechanisms overlap, ensuring different molecule shielding and metabolic reprogramming and supporting glass formation by different molecules and water replacement. The latter is also assisted by non-protein molecules such as trehalose, although not all tardigrades rely on this disaccharide [16,31,42,43]. Further study of other non-protein protectants may provide additional useful information concerning dehydration tolerance in anhydrobiotic tardigrades. The same applies to TDP and Dsup; the results of multiomic studies indicate different numbers of paralogs for these proteins [28] and a lack of conservation of these proteins between Eutardigrada and Heterotardigrada but also a possibility of convergent evolution of anhydrobiosis machinery [29,30,44,45]. However, the role of highly conserved and ubiquitous heat shock proteins (HSPs) in managing different kinds of cellular stress and providing proteostasis is not consistent in the case of tardigrade anhydrobiosis [46–48]. With contradictory data, the role of these proteins in different anhydrobiotic tardigrades remains to be verified.

Damage caused by oxidative stress appears to be the most deleterious effect of water depletion, mediated by the formation of reactive oxygen species (ROS) [49,50]. ROS are involved in many pathological processes, including aging [51,52]. Genomic-, transcriptomic-, and proteomics-based studies have indicated the expression of a wide variety of known antioxidant enzymes in dehydrated tardigrades compared to active ones [30,53–55]. These enzymes can limit the availability of ROS and include superoxide dismutase (SOD), which transforms superoxide anions into hydrogen peroxide ($H_2O_2$); catalase (CAT) and glutathione peroxidase (GPx), which decompose $H_2O_2$ and glutathione transferase (GST), catalyzing the detoxification of endogenously derived ROS (and environmental pollutants) by glutathione conjunction; and glutathione reductase (GR), which recycles glutathione from glutathione disulfide [41,54,56]. Duplication of SOD-encoding genes was

observed as a common characteristic of anhydrobiotic tardigrades [28,30]. Additionally, in *Paramacrobiotus richtersi* (Murray, 1911), increased SOD activity was reported in response to dehydration, suggesting its importance in the process [54]. Moreover, upregulation of catalase-encoding genes during anhydrobiosis was detected in the tardigrade *Hypsibius exemplaris* Gąsiorek, Stec, Morek, & Michalczyk, 2018 [29]. Glutathione peroxidase was reported to be crucial for successful anhydrobiosis in *Pam. spatialis* Guidetti, Cesari, Bertolani, Altiero, & Rebecchi, 2019 [41]. However, despite the available data concerning antioxidant systems in anhydrobiotic tardigrades, the molecular mechanism underlying anhydrobiosis is not completely understood. Further studies are necessary to understand the role of antioxidant systems in anhydrobiotic species.

## 3. Aging

Aging is a universal process that can be defined as the progressive decline in biological functions leading to increased vulnerability to disease and death [57–60]. Aging is associated with decline in behavioral (e.g., alarm reaction and sensitivity to conditioning), life-history (e.g., lifespan and fecundity), morphological (e.g., body size and body shape), and physiological traits (e.g., oxygen consumption and resistance to stress) [56]. There is great diversity in aging rates among species, geographical populations, and individuals within species [61,62]. Moreover, not all tissues and organs age at the same rate [63]. It is assumed that a variety of aging rates has evolved to meet the challenges of specific environments; understanding the underlying adaptations can provide valuable insights into aging [64], with the possibility of counteracting human physical and cognitive disability [62].

### 3.1. Levels of Research

It is known that aging is not dependent on a single gene and is a consequence of multiple processes that may interact and operate at different levels of functional organization [65]. Several explanations of aging have been proposed that focus on different levels of organization, programmed or adaptive aging theories (evolutionary theories) and mechanistic or damage theories (molecular, cellular, and systematic theories) [66,67].

Human aging research is difficult for many reasons, including ethical issues, a long natural lifespan, environmental influences, and demographic variability. Several animal models, including the nematode *Caenorhabditis elegans* (Maupas, 1900), fruit fly (*Drosophila melanogaster* Meigen, 1830) and rodents, and single-cell organisms such as yeast (*Saccharomyces cerevisiae* Meyen ex E.C. Hansen) have been developed to study the fundamental aspects of aging biology [63,68]. These organisms offer certain experimental advantages that make them suitable models for the study of aging. For instance, *C. elegans*, *D. melanogaster*, and *S. cerevisiae* have shorter lifespans and are easy to handle and culture [69], whereas rodents have a closer genetic proximity to humans [70] and can be genetically manipulated and phenotypically characterized [71]. Studies on such models have produced useful insights concerning molecular and cellular mechanisms underlying aging and appear to demonstrate the complexities of the process at higher levels of organization [68,70,72]. Thus, using these models to verify aging theories and in experimental examination of age-related diseases may reveal hidden aspects of aging biology.

Studies on the model organisms have identified several extracellular and intracellular aging hallmarks [73]. The models also provide the possibility of the hallmarks impacting research at higher orders of phenotype complexity, including organism morphology, physiology, and behavior [74]. Moreover, the models are useful in evolutionary studies based on life-history theory and for the explanation of variations in the timing of fertility, growth, developmental rates, and death of living organisms [75]. This requires an understanding of the life-history of an organism, defined as its pattern of survival and reproduction, along with the traits that directly affect survival and the timing and amount of reproduction. The traits include: (1) growth rate; (2) age and size at sexual maturity; (3) the temporal pattern

or schedule of reproduction; (4) the number, size, and sex ratio of offspring; (5) mortality rates; and (6) patterns of dormancy and dispersal [76].

Years of research on aging did not produce a complete understanding of the mechanism(s) of the process. An integrated multidisciplinary and multilevel approach using novel model organisms may contribute to a better outcome.

*3.2. Proposed Mechanism*

At the population level, aging manifests as a reduction in survival and fecundity in later stages of adulthood [77]. According to the evolutionary theory of aging, there are two approaches to explain why aging originated and is maintained in populations. The first concentrates on phenotypic life-history traits (Section 3.1) and consequences of reproductive costs; the second focuses on genetic effects that arise when natural selection pressures decrease with age [77,78]. The life-history theory claims to explain how the environment affects the survival and reproduction of organisms at different ages and how life-history traits are connected to each other [79]. Accordingly, many studies have been conducted to determine the trade-off between lifespan and fecundity and their roles in the aging process (e.g., [80–82]). Two sub-types of natural selection pressure are distinguished. The first suggests that aging is a byproduct of selection of other beneficial traits; the second suggests that natural selection is unable to prevent deterioration of older organisms because it attenuates with age. However, both sub-types embrace the common principle that natural selection affects lifespan and/or reproduction [83].

The cellular mechanism of aging was proposed by L. Hayflick in 1985, known as the telomere theory of aging [84]. It assumes that the process of cell senescence limits the number of cell divisions and can occur with the arrest of cell proliferation (replicative senescence) or from other causes (stress-induced senescence) [65]. Replicative senescence results mainly from telomere shortening [67], whereas stress-induced senescence occurs in response to stressors such as oxidative stress, mitogenic stress resulting in DNA damage, changes in heterochromatin structure, and other cellular changes, including strong mitogenic signals resulting from oncogene expression [85]. The theory assuming cellular senescence is compatible with various "accumulation theories", including free radical theory and the somatic mutation theory of aging. All living organisms produce free radicals along with ROS, which originate mainly from mitochondria and cause oxidative damage to cellular molecules, including DNA [66,86]. The somatic mutation theory proposes that mutations accumulating in cells are specific causes of senescence and that oxidative stress caused by ROS is an important cause of damage [66]. Additionally, some studies have focused on specific gene variants associated with longevity (e.g., [87]).

Studies at molecular and cellular levels have revealed several gene-mediated phenomena contributing to aging; some are categorized as key aging hallmarks, including genomic instability, telomere shortening, epigenetic alterations, deregulated nutrient-sensing, mitochondrial dysfunction, cellular senescence, stem cell exhaustion, and altered intracellular and intercellular communication [88]. The number of studies identifying "longevity genes" has increased in recent decades (e.g., [89,90]). Anhydrobiosis appears to increase lifespan [17,20], but few studies support this. Thus, an approach combining aging hallmarks and identified "longevity genes" in the context of anhydrobiosis may uncover hidden aspects of aging mechanisms, which may validate aging theories and the "Sleeping Beauty" hypothesis.

## 4. Markers in Research on the Effect of Anhydrobiosis on Aging

The identification of possible markers of tardigrade aging will allow the study of anhydrobiosis as an anti-aging strategy (Figure 1). Available data indicate that the markers may be identified from an analysis of life-history traits and cellular processes correlated with aging. However, the use of markers in studies on the anhydrobiosis effect on tardigrade aging requires data on the efficiency of anhydrobiosis for the species in the study. Moreover,

it should be considered that anhydrobiosis survival and recovery rate may depend on tardigrade age.

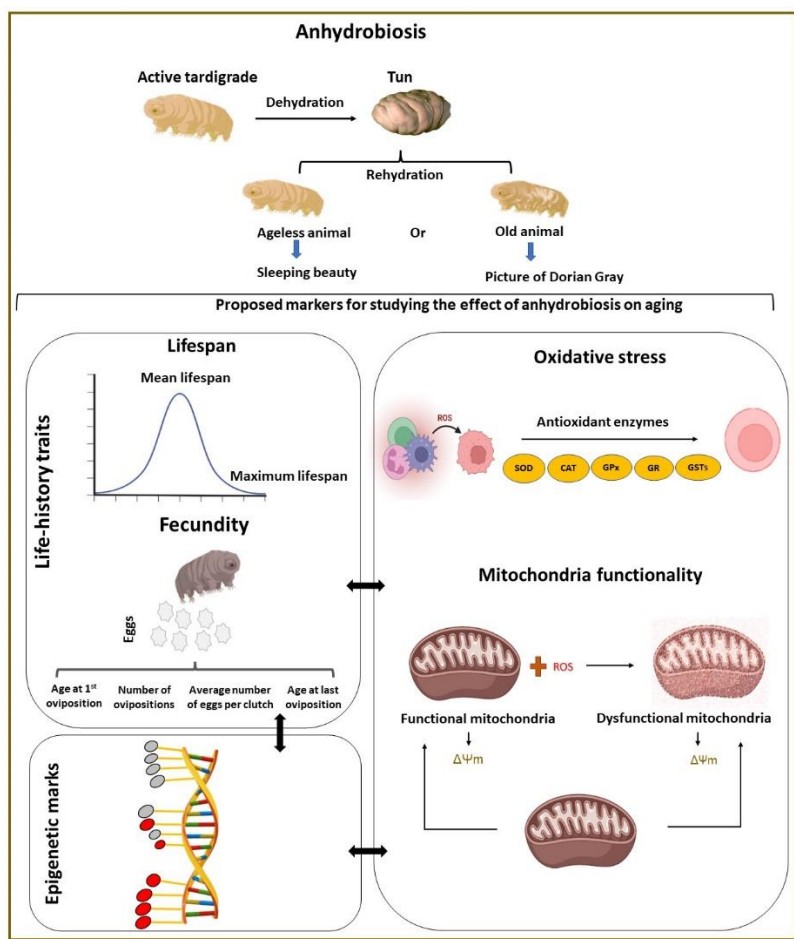

**Figure 1.** Life-history and cellular traits for studying the effect of anhydrobiosis on aging. ROS: reactive oxygen species; SOD: superoxide dismutase; CAT: catalase; GPx: glutathione peroxidase; GR: glutathione reductase; GSTs: glutathione S-transferase; $\Delta\Psi$m: mitochondrial membrane potential.

*4.1. Life-History Traits*

Life-history traits consider the lifespan of an organism, the way it develops, reproduces, and dies [91]. For tardigrades, the following traits are considered: lifespan, number of eggs (fecundity), number of molts, clutch size, hatching success, hatching percentage, age at first oviposition, and total number of ovipositions [92,93], although their use in studying the effect of anhydrobiosis on aging is rare. The traits most often used are lifespan and fecundity (Figure 1) [17,20], both widely used in aging studies on other organisms (e.g., [80,82]). The traits for different tardigrade species and the species anhydrobiosis efficiency are summarized in Table 1. However, the conditions in these studies, including diet, ambient temperature, relative humidity, water quality, and the culture substratum, can impact the life-history traits. The same applies to anhydrobiotic success, as it can be affected by the culture conditions and other factors, such as overall body size, temperature, the mode of dehydration, and duration in the tun state, e.g., [8,16,17,37].

**Table 1.** Life-history traits in tardigrade species. ALS: average lifespan; ML: maximum lifespan; AFO: age at first oviposition; NO: number of ovipositions; ANEC: average number of eggs per clutch; ALO: age at last oviposition; SSLH: sample size in study of life-history traits; ND: no data; * = SD values are not reported; tardigrade genera abbreviations according to Perry et al. [94,95].

| Species and Reproduction Mode | Lifespan | | Fecundity | | | | SSLH | Anhydrobiosis Capability | References |
|---|---|---|---|---|---|---|---|---|---|
| | ALS (Days) (Mean + SD) | ML (Days) | AFO (Days) (Mean + SD) | NO (Mean + SD) | ANEC (Mean + SD) | ALO (Mean ± SD) | | (High/Medium/Low) | |
| *Acu. antarcticus* generation F1; parthenogenesis | 88.8 ± 20.0 | ND | 17.1 ± 3.6 | 6.4 ± 1.0 | 1.8 ± 0.8 | 22 * | 22 | ND | [96] |
| *Acu. antarcticus* generation F2; parthenogenesis | 49.5 ± 26.4 | ND | 16.9 ± 3.5 | 3.0 ± 1.9 | 1.7 ± 0.8 | 30 * | 43 | ND | [96] |
| *Acu. antarcticus*; parthenogenesis | 69.2 ± 36.4 | ND | 9.3 ± 1.1 | 7.5 ± 4.3 | 3.4 ± 2.6 | 66 * | 68 | High | [97] |
| *Dip.* cf. *scoticum*; parthenogenesis | ND | 263 | ND | 15 | 1.3 ± 4.7 | ND | 1 | ND | [98] |
| *Gre. myrons*; parthenogenesis | 18.8 ± 7.0 | 30 | ND | ND | ND | ND | 29 | ND | [93] |
| *Hys. exemplaris*; parthenogenesis | 61.9 ± 9.9 | 75 | 8.0 ± 3.1 | 10.5 ± 2.2 | 8 * | ND | ≈100 | Low | [99] |
| *Iso. dastychi*; dioecious | ND | 29 | ND | ND | ND | ND | 15 | ND | [100] |
| *Mac. hefelandi*; parthenogenesis | ND | 84 | 31 | ND | ND | ND | ND | High | [101] |
| *Mac. joannae*; dioecious | ND | 266 | ND | ND | 1.7 ± 5.3 | ND | ND | ND | [98] |
| *Mac. sapiens*; dioecious | 83.0 ± 33.5 | 145 | 16.5 ± 3.8 | ND | Max no 16 * | ND | 66 | ND | [6] |
| *Meb. joenssoni*; dioecious | 86.5 ± 18.0 | 150 | 19.8 ± 1.7 | ND | ND | ND | 224 | ND | [102] |
| *Mil. tardigradum*; parthenogenesis | 82.7 ± 2.7 | 107 | ND | ND | ND | ND | 16 | High | [103] |
| *Mil. tardigradum*; parthenogenesis | 42.7 ± 11.8 | 58 | 15.3 ± 1.6 | 12 * | 1.8 ± 11.2 | ND | ND | High | [16,104] |
| *Pam. fairbanski* clone 1 parthenogenesis | 194.9 ± 164.4 | ND | 76.9 ± 16.4 | ND | ND | ND | 16 | ND | [105] |
| *Pam. fairbanski* clone 2 parthenogenesis | 137.3 ± 136.4 | ND | 70.7 ± 19.4 | ND | ND | ND | 15 | ND | [105] |
| *Pam. kenianus*, Population (I); parthenogenesis | 125 ± 35 | 204 | 10 * | ND | ND | ND | 22 | ND | [92] |
| *Pam. kenianus*, population (II); parthenogenesis | 141 ± 54 | 212 | 10 * | ND | ND | ND | 22 | ND | [92] |
| *Pam. palaui*; parthenogenesis | 97 ± 31 | 187 | 10 * | ND | ND | ND | 22 | ND | [92] |
| *Pam. richtersi*; parthenogenesis | ND | ND | 64.2 ± 1.7 | ND | ND | ND | 80 | High | [106] |
| *Pam. tonollii*; dioecious | 69.0 ± 45.1 | 237 | 24.4 ± 4.4 | ND | Max no 19 * | ND | 104 | ND | [6] |

**Table 1.** *Cont.*

| Species and Reproduction Mode | Lifespan | | | Fecundity | | | SSLH | Anhydrobiosis Capability | References |
|---|---|---|---|---|---|---|---|---|---|
| | ALS (Days) (Mean + SD) | ML (Days) | AFO (Days) (Mean + SD) | NO (Mean + SD) | ANEC (Mean + SD) | ALO (Mean ± SD) | | (High/Medium/Low) | |
| *Ram. oberhaeuseri;* parthenogenesis | ND | 70 | ND | ND | ND | ND | ND | High | [107,108] |
| *Ram. varieornatus;* parthenogenesis | 13-87 | 87 | ND | ND | ND | ND | 10 | High | [109] |

Note: Due to lack of sufficient data on life-history traits, several tardigrade species, *Acu. antarcticus* (**generation P**), *Hab. crispue, Hys. convergens, Not. arcticus Ram. subanomalus, Ech. trisetosus, Ecn. sigismundi, Ric. coronifer Dip. chilenense, Ech. jenningsi, Ech. testudo, Meb.furciger, Mac. areolatus, Mil. inceptum*, and *Pam. Spatialis*, are omitted from the table. These species differ in anhydrobiotic ability. *Dip. chilenense, Ech. jenningsi, Ech. testudo, Meb.furciger, Mac. areolatus, Mil. inceptum*, and *Pam. spatialis* exhibit high anhydrobiotic ability; *Ech. trisetosus, Ecn. sigismundi*, and *Ric. coronifer* are reported to exhibit moderate anhydrobiotic ability. F1 and F2 for *Acu. antarcticus* denote all offspring generated from two females belonging to a parental generation of adult females. The discrepancy in lifespan determined for F1 and F2 probably reflects phenotypic plasticity of life-history traits proposed as a possible survival strategy in the colonizing habitats of many invertebrates subjected to extreme and stochastic environmental conditions [95].

### 4.1.1. Lifespan

The maximum lifespan or total lifespan refers to the age at which the oldest member of the species or experimental group died [110]. The maximum lifespan often differs from the mean or average lifespan and longevity. The mean lifespan is a statistical measure of the average time an organism is expected to live and corresponds to life expectancy. The maximum lifespan is usually determined by the rate of aging, whereas mean lifespan varies with susceptibility to injury and disease [111]. The term "active lifespan" is generally connected to the amount of time spent in an active state, while the latent state may be represented by anhydrobiosis [112]. Longevity does not refer to the specific maximum lifespan; it refers to the relatively long lifespan of some members of a population [113]. Increased longevity directly correlates to extension of the maximum lifespan of an individual [113]. For the effect of anhydrobiosis on aging, increased longevity corresponds to the cumulative duration spent in the tun stage [17]. Accordingly, the estimation of maximum lifespan is a prerequisite to determine whether aging occurs in the tun stage. The same applies to mean lifespan, as it helps in estimating the average age at which a member of a population will die. Thus, the estimation of maximum or mean lifespan is necessary to address anhydrobiosis effects on aging.

It is known that the maximum and mean lifespan in tardigrades vary between species. The parameters are scored by estimating specimen activity, including food intake and mobility. To ensure that tardigrades are dead and not in a cryptobiotic or quiescent state, the latter caused by molting or preparing for egg laying, immobility is considered with the appearance of a straight and transparent body, although fluorescent dyes specific to dead cells are also proposed to be used [9,114]. The maximum lifespan in tardigrades is suggested as 1–24 months (excluding the period of cryptobiosis); the mean lifespan is 19–195 days [7,115]. The longest maximum lifespan was recorded for *Halobiotus crispae* Kristensen, 1982 (730 days), and the shortest maximum lifespan was recorded for *Grevenius myrops* (du Bois-Reymond Marcus, 1944) (30 days) [93,116]. The longest mean lifespan was recorded for *Pam. fairbanski* (clone 1) (194.9 ± 164.4 days); the shortest mean lifespan was recorded for *Gre. myrops* (18.8 ± 7.0 days) [93,104]. The maximum and mean lifespan of other species are presented in Table 1. Lifespan (mean or maximum) is diverse throughout the tardigrade lineage. The same applies to anhydrobiotic capability. Species with a longer lifespan do not necessarily exhibit greater anhydrobiotic capability than those with a shorter lifespan. Tardigrades with a longer lifespan require more time to grow. These data are important in determining suitable species for studying the effect of anhydrobiosis on aging.

### 4.1.2. Fecundity

Fecundity patterns reflect the physiological state of individuals, usually corresponding to their age [117]. The term generally refers to the total number of offspring in a particular time period [118]. Parameters associated with fecundity include number of eggs, number of reproductive days, reproductive effort, and age at first and last reproduction. These parameters are reported to be closely related to longevity [119]. Thus, fecundity and related parameters appear to be crucial for the study of anhydrobiosis in tardigrade aging.

Most marine species of tardigrades are dioecious (both female and male), whereas parthenogenesis (a self-fertilization strategy) is most common among limno-terrestrial species, although some limno-terrestrial species are also dioecious or hermaphroditic (have both types of reproductive organs) [5]. Eggs are the product of different modes of fertilization (internal and external). The total number of eggs resulting from the overall reproductivity of an individual determines the total number of offspring. The first appearance of eggs in a female ovary is considered to be an indication of sexual maturity [6]. The available food source, temperature, parasites, and number of animals in the surrounding environment directly affect egg production [98,102]. Two egg deposition patterns are observed for tardigrades; they lay eggs freely in the surrounding environment or in exuviae [120]. The number of eggs can vary in the same species, depending on age [121]. The egg-laying patterns of many tardigrade species (especially marine species) are unknown. Information on the reproduction mode and fecundity patterns is important in determining suitable species for the study of anhydrobiosis in aging. The identification of males and females in a species, time of sexual maturity (first oviposition), total number of ovipositions, and average number of eggs per clutch are some of the crucial parameters.

Unfortunately, data on changes in fecundity and other reproductive parameters over the lifespan of tardigrade species are still limited. This data could be directly linked to the understanding of species-specific or population-specific characteristics of reproduction [97]. For instance, an increase in oviposition intervals was observed to be directly correlated with lifespan in *Acutuncus antarcticus* (Richters, 1904) [97], suggesting that reproductive senescence is associated with aging. Thus, the frequency of oviposition events and age at the last oviposition could be putative indicators of aging in an organism and possible markers for studying the effect of anhydrobiosis on aging.

### 4.2. Possible Cellular Markers

Notwithstanding recent advancements in the understanding of mechanisms governing the aging process [74,122,123], the biological basis and factors associated with the process remain somewhat unclear. At the cellular level, aging is a process driven by accumulation of irreversible molecular and cellular damage resulting in a risk of functional decline, disease, and, ultimately, death [124]. The aging process is not dependent on a single gene and is a consequence of processes that may interact and operate at many levels of functional organization (Section 3). It is also possible that cumulative damages caused by stochastic, environmental, and genetic factors are the main drivers of lifespan variation and aging patterns [124]. The identification of molecular and cellular mechanisms underlying tardigrade aging may allow an indication of markers suitable for research on the effect of anhydrobiosis on aging.

### 4.2.1. Oxidative Stress

In 1928, R. Pearl proposed that lifespan is inversely related to metabolic rate [125]. In 1956, D. Harman proposed the free radical theory of aging (later known as the oxidative stress theory of aging), essentially a biochemical explanation of the theory proposed by R. Pearl [126]. According to the free radical theory, free radicals (including highly reactive derivatives of oxygen, ROS) are produced during normal metabolism. Over time, an organism becomes unable to neutralize the damage that they cause. These damages accumulate with time and threaten the homeostasis of the organism, accelerating aging and ultimately leading to death [127]. It has been assumed that a higher rate of metabolism

results in greater production of free radicals and, consequently, in faster aging and reduced lifespan. Some experiments did not confirm the simplified relationship between metabolism rate and aging [128–130]; others supported this assumption. Damage to DNA, proteins, and lipids has been shown to increase with age in many organisms, including humans, mice, flies, and *C. elegans* [130]. Reduced antioxidant defenses and increased oxidative stress have been shown to reduce lifespan [131]. Taken together, it is reasonable to consider oxidative stress and related factors in search of possible markers for studying the effect of anhydrobiosis on aging, although oxidative stress is also related to anhydrobiosis [41,103, 132]. It is proposed that the term "oxidative stress" be replaced with the term "oxidative signaling" [133].

ROS Generation and Oxidative Modifications of Different Molecules

It is commonly accepted that excessive formation of ROS and limited anti-oxidative defenses cause imbalances resulting in deleterious damage [134]. In normal metabolism, generation and elimination of ROS are balanced by antioxidant mechanisms (Figure 1) [135]. However, in stress conditions, greater ROS production occurs that cannot be removed by antioxidant defense, resulting in oxidative damage to key molecules and oxidative stress enhancement [103]. The molecular mechanisms underlying the anti-oxidative defense and the role of ROS in the biology of aging and in the development of age-related diseases remain somewhat unclear. ROS are characterized as a variety of molecules including free radicals (chemical species with one unpaired electron) derived from molecular oxygen, such as the superoxide anion ($O_2^{\bullet-}$), the hydroxyl radical ($HO^{\bullet}$), and hydrogen peroxide ($H_2O_2$) [136]. The main source of ROS is electron transport during ATP synthesis in mitochondria [137]. Studies on the impact of oxidative stress on anhydrobiotic tardigrades are limited. However, a study on the tardigrade *Mil. tardigradum* indicated a relationship between DNA damage and the duration of the tun stage; damage increased with the duration of the desiccated state [138]. Nevertheless, it has not been explained yet whether the DNA damage, known to be enhanced during aging, can be affected by the age of tardigrades. However, it has been shown that, in *Pam. spatialis*, ROS production significantly increases as a function of time spent in anhydrobiosis [41]. Another study on *Pam. richtersi* demonstrated that heat stress, related to oxidative stress, resulted in greater DNA damage in tuns [139]. Oxidative conditions are thought to be responsible for the death of anhydrobiotic tardigrades and for a longer required recovery time for repair of oxidative damages [139]. All data support the assumption that successful anhydrobiosis is based on effective anti-oxidative defense. Accordingly, the weakening of oxidative conditions (e.g., by application of low temperature) helps to prevent oxidative damage and suppress aging [140]. Thus, mechanisms of oxidative stress and anti-oxidative defense can be used to determine markers suitable for studies on anhydrobiosis impact on aging.

It is known that ROS oxidize all types of cellular components; the process is known as oxidative modification. This modification may be crucial for normal cell functioning [141]; however, when it prevents molecules from performing their native functions, cellular dysfunction results [133]. The modification may concern DNA, but it is also crucial for protein functioning. The oxidative modifications of proteins are characterized mainly by the addition of a carbonyl group. Increased carbonyl-bearing proteins appear to be connected with the aging process [133]. Additionally, increased protein carbonylation has been detected in a UV-stressed tardigrade *Hys. exempleris* [142]. Furthermore, oxidation of cysteine residues has been found to be linked with anhydrobiosis in *Mil. tardigradum* [143].

Another important oxidative modification is known as "lipid peroxidation". Fatty acids are a major source of energy in the cell. The composition and structure of fatty acids affect the potential for oxidative damage including peroxidation [144]. Fatty acids contain at least four carbon atoms, with a carboxylic group at the end of the molecule. They are divided into three main groups on the basis of their carbon–carbon bonds: saturated fatty acids (SFAs—single carbon–carbon bonds), monounsaturated fatty acids (MUFAs—one carbon–carbon double bond), and polyunsaturated fatty acids (PUFAs—two or more

carbon–carbon double bonds). PUFAs are abundant in cellular/organelle membranes and especially susceptible to ROS-induced peroxidation [145]. The most commonly used markers of fatty acid peroxidation are thiobarbituric acid reactive substances (TBARS), hexanoyl lysin (HEL), and total antioxidant capacity (TAC) [145]. Studies related to lipid peroxidation in tardigrades are rare. Increased TBARS were reported in anhydrobiotic *Pam. richtersi* [54]; however, the impact on lifespan and aging were not discussed.

Verification of DNA, protein, and fatty acid modifications over the lifespan appears reasonable in searching for indicators of aging in tardigrades and the role of anhydrobiosis in the process.

ROS-Scavenging Enzymes

In the course of evolution, several powerful anti-oxidative mechanisms have developed to protect against oxidative stress [146], taking advantage of endogenous antioxidant enzymes, including SOD, CAT, GPx, and GR, that have the ability to decompose ROS, providing protection against oxidative modification (Section 2 and Figure 1). Their effectiveness may vary with the physiological stages of the organism, including age [147], resulting in changes in lifespan and physiological functions [148]. There is also evidence not supporting any consistent relationship between age-related changes in antioxidant enzyme activities and lifespan [149].

In tardigrades, the activity of these enzymes during anhydrobiosis has been described (Section 2). In the tardigrade *Pam. spatialis,* GPx is reported to be critically important for desiccation tolerance. In the same study, GR and CAT are also highlighted as key components during the rehydration stage [41]. Additionally, the upregulation of two proteins, glutathione S-transferase (GST) and pirin-like protein, has been reported in response to desiccation in the tardigrade *Hys. exempleris*, indicating their role in the strategy against oxidative stress [150]. A novel manganese-dependent peroxidase (g12777) was identified as an important factor for anhydrobiosis survival in the tardigrade species *Ram. varieornatus* Bertolani & Kinchin, 1993 [149]. Recently, CAT and SOD activities were reported to be crucial for successful anhydrobiosis in *Pam. spatialis* and *Acu. antarcticus*, respectively [132]. An analysis of antioxidant enzyme performance in tardigrades, particularly in animals of different ages, may help in the verification of the enzymes as possible markers of the anhydrobiosis effect in aging.

Mitochondria Functioning

The mitochondrion is the center of cellular metabolism that controls many cell functions by mechanisms enabling the fine-tuning of gene expression levels by intracellular reduction–oxidation state and metabolite-derived nuclear epigenetic marks [151]. Thus, it is commonly accepted that aging is associated with a decline in mitochondrial functions [152]. As the mitochondrial respiratory chain is the major source of ROS (ROS Generation and Oxidative Modifications of Different Molecules), mitochondria are susceptible to excessive oxidative damage (Figure 1) [153], although they contain antioxidant enzymes, including SOD, CAT, and GPx [154]. Oxidative damage to mitochondrial DNA (mtDNA) increases its mutation rate [155], contributing to the pathophysiology of age-associated diseases, with changes in mtDNA copy number reported to coincide with aging. Impairment of mitochondrial energy transformation dysregulates nutrient sensing; alters epigenetic mechanisms; and causes a decline in cell functions [156].

Animal mitochondria contain over 1000 proteins encoded mainly by the nuclear genome. Thus, efficient communication between mitochondria and the nucleus (mitonuclear communication) is necessary to maintain correlative responses in a constantly changing intrinsic and extrinsic cellular environment [157]. Accordingly, impaired mitonuclear communication is reported to be strongly related with aging and age-related diseases [158]. Additionally, mitochondria are known to play an important role in biotic stress responses [159].

Data concerning the involvement of mitochondria in tardigrade aging are not available. However, in tardigrades, mitochondria appear to play an important role in anhydrobiosis, because uncoupling (elimination of coupling between electron transport and ATP synthesis) of mitochondria suppresses tun formation. Moreover, the organelles contribute to tun functionality and successful rehydration [24,160]. *Hys. exempleris* tun degeneration has been shown to occur with changes in mitochondrion ultrastructure [161]. Moreover, mitochondrial alternative oxidase (AOX) activity has been shown to be important for *Mil. inceptum* revival from the long-term tun stage [27]. Nevertheless, the role of mitochondria in tardigrade anhydrobiosis is still not completely understood. Studies on selected mitochondrial markers may produce a better understanding of the anhydrobiosis process and the impact of anhydrobiosis on tardigrade aging. The markers may include mtDNA copy number, metabolome changes important for epigenetic mechanisms, mitochondrial inner membrane potential, and selected mitochondrial proteins, including RvLEAM (*Ram. varieornatus* mitochondrial late embryogenesis abundant), MAHS (mitochondrial abundant heat soluble), and AOX. RvLEAM and MAHS proteins are described as potential mitochondrial protectants during anhydrobiosis [162], and AOX contribution to the state is mentioned in the Introduction.

### 4.2.2. Epigenetic Modifications

Epigenetics refers to reversible heritable mechanisms that affect gene expression via chromatin modifications, causing changes in DNA availability [163]. Chromatin is the polymer of nucleosomes composed of DNA and histone proteins. A single nucleosome contains a histone octamer consisting of two copies of four different histones (H2A, H2B, H3, H4) or the histone variants (e.g., macroH2A, H3.3, and H2A.Z). The octamer is wrapped by a short fragment (147 base pairs) of DNA [164] controlled by histone H1 [165], known as a linker histone, as it links adjacent nucleosomes [166]. Histones are highly conserved, basic proteins abundant in lysine and arginine residues that can be modified. Notably, both histones and DNA are modified, affecting gene expression [167].

Previous studies indicate a link between epigenetic mechanisms and aging. The loss of histones and chromatin remodeling resulting in transcriptional changes are key epigenetic hallmarks of aging [168]. Moreover, the relationship between mitochondrial functioning and epigenetic mechanisms of nuclear gene expression regulation is emerging in terms of the role of mitochondria in health and aging [169]. It should be remembered that epigenetic alterations also concern mitochondrial DNA and nucleoid proteins (as mitochondria lack histones) that may be directly linked to the stress response and the aging process, although it is not fully understood [170,171]. Studies on epigenetic mechanisms including mitochondria may provide important insight into the impact of anhydrobiosis on aging.

### Epigenetics and Stress Conditions

Exposure of organisms to environmental stress can affect the fitness of their offspring for generations [20]. The effect of parental exposure to environmental stress and its transmission to offspring is known as inter-generational inheritance or epigenetic inheritance [172]. It was reported to be caused by the transmission of epigenetic markers from one generation to another, resulting in changes in traits of offspring [170]. They include histone modifications, altered expression of micro-RNAs, DNA modifications, and relevant changes in the activity of enzymes controlling epigenetic modifications [173]. Epigenetic inheritance appears to be an important issue in research on the role of the epigenetic mechanism in aging. The same applies to anhydrobiosis, which represents core survival strategies to survive harsh conditions. Therefore, studies related to epigenetic mechanisms may produce markers useful in the study of the impact of anhydrobiosis on aging.

Possible Epigenetic Markers

DNA methylation is one of the best known epigenetic modifications [174]. It is catalyzed by DNA methyltransferases (DNMTs) responsible for the formation of 5mC (5-methylcytosine) [174]. Compared with vertebrate DNA, invertebrate DNA is sparsely methylated [175]. Nevertheless, in vertebrate models, the stress response is often accompanied by DNA methylation and other chromatin modifications [176,177]. Although one DNA methyltransferase (DNMT2) has been identified in tardigrades [178], its role in anhydrobiosis or aging has not been studied. DNA methylation or DNMT2 activity may be reasonable putative markers in the study of the impact of anhydrobiosis on aging.

Histone modifications are more diverse. The most common are the methylation of arginine or lysine residues or acetylation of lysine residues. Methylation of histones can affect other proteins (transcription factors), whereas acetylation partially unwinds DNA, making it more accessible for gene expression [179]. The role of histone modifications in the aging process has been extensively investigated; however, whether these modifications are causes or effects of aging is uncertain [180]. It has been reported that alternations in histones, such as trimethylation of lysine 20 on histone H4 (H4K20me3), trimethylation of lysine 9 on histone H3 (H3K9me3), trimethylation of lysine 27 on histone H3 (H3K27me3), and acetylation of lysine 9 on histone H3 (H3K9ac), are associated with aging [179]. Moreover, combined effects of histone modifications appear to be essential for regulation of stress-responsive gene expression [181].

Epigenetic studies in tardigrades are rare; however, histone H1 has been identified in *Ram. varieornatus*, and its influence on chromatin structure has been discussed [182]. Histones H4 and H2B.2 were identified in *Mil. tardigradum* during the dormancy state [143], but their role in anhydrobiosis was not reported. Studies on histones and on histone and DNA modifications may produce possible markers to study the effect of anhydrobiosis on aging. Such studies could be supplemented by miRNAs, reported to be involved in cellular stress responses and in lifespan regulation in several organisms [183], as proper data are available for tardigrades, e.g., [29].

**5. Conclusions**

The impact of anhydrobiosis on aging has not been widely investigated and is not completely understood. Most previous studies have explained the extraordinary capacity of anhydrobiotic animals to survive extreme conditions; however, few data are available concerning their aging patterns and underlying processes. It is also not clear whether anhydrobiosis affects aging or suppresses possible causes of death. Most invertebrate aging research has been conducted using limited animal models; there are many opportunities for tardigrades in studies of aging biology, including molecular, cellular, and life-history traits. These studies may contribute to the verification of aging theories and hypotheses such as the "Sleeping Beauty" and "The Picture of Dorian Gray", with important applicative consequences. They may concern medicine, biotechnology, and astrobiology and result in improved anti-aging strategies and preservation of biological materials for transplantation or pharmaceutical products and dry foods.

**Author Contributions:** All authors contributed to writing and accepted the final version of the manuscript. All authors have read and agreed to the published version of the manuscript.

**Funding:** Amit Kumar Nagwani is a scholarship passport holder of the Interdisciplinary Doctoral Studies at the Faculty of Biology, Adam Mickiewicz University, POWR.03.02.00-00-I006/17. This research was supported by research grants of the National Science Centre, Poland, NCN 2016/21/B/NZ4/00131 and 2021/41/N/NZ3/01165.

**Institutional Review Board Statement:** Not applicable.

**Data Availability Statement:** Not applicable.

**Acknowledgments:** The studies were performed partly in the framework of the activities of BARg (Biodiversity and Astrobiology Research group).

**Conflicts of Interest:** The authors declare no conflict of interest.

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
