# Peer review of "Applicable Life-History and Molecular Traits for Studying the Effects of Anhydrobiosis on Aging in Tardigrades"

_diversity, doi:10.3390/d14080664_

Round 1

Reviewer 1 Report

This review article by Nagwani et al. summarizes progress and open opportunities in the field of tardigrade anhydrobiosis and aging. They draw particular attention to the unresolved question of whether or not tardigrades are aging during anhydrobiosis or cryptobiosis. The two competing hypotheses of the “Sleeping Beauty” and “Picture of Dorian Gray” are discussed. This review is an appropriate call to action for rigorous studies in this area of research. The article would benefit greatly from a reorganization to keep the central question of the relationship between aging and anhydrobiosis in focus throughout the writing.

Major Comments

When first introducing the competing hypotheses of aging during cryptobiosis it would be helpful to spell out in more detail the conditions for each, emphasizing whether or not time spent in cryptobiosis counts against lifespan upon recovery, as well as the signs of aging and/or damage that may or may not occur during cryptobiosis. The metabolic features of each model should be discussed here as well. When listing the species that were suggested to follow the sleeping beauty or Dorian Gray hypotheses, it is not clear what the conditions are for these classifications. (Lines 58-64)

Table 1 is a nice summary of lifespan and fecundity data from the literature. Would it be possible to indicate in the table the number of animals used in each of these studies (the error on some like Pam. fairbanski are huge)? The legend says ANEC, but the table says NEC. What is the difference between F1 and F2 for Acu. Antarcticus and why is there such a discrepancy in lifespan? I believe the references are not correct for much of the data in the table – these should be checked and corrected to make sure they correspond to the appropriate studies.

The role of environmental factors like food, temperature, water quality, etc. can have a significant effect on phenotypic outcomes like lifespan and fecundity. The role of these should be mentioned and perhaps explained as a caveat for the reported data summarized from all the studies in table 1. For example, with differing diets and culture conditions clutch sizes could change significantly and lifespan and other traits could similarly be impacted. Alerting readers to this seems important.

There are reports of DNA damage occurring during anhydrobiosis (for example, Neumann et al 2009 – this is currently reference 102) that could be included in the discussion about DNA damage as a possible marker for assessing aging during anhydrobiosis.

It does not entirely make sense to include a section (4.2.1.1) on ROS after saying that the ROS theory of aging has generally been shown to not be correct.

An emphasis in the epigenetics section is on histone modifications that impact chromatin accessibility. There is then the claim that mitochondria are also subject to epigenetic regulation – yet, they lack histones. The connection here is not clear. I recommend significantly truncating if not removing the section on epigenetics because the majority of this section reviews the basic biology of epigenetic inheritance with no reference to tardigrades. Only the last paragraph is relevant and correctly states that there is little to no work that has been done on this topic. It is intriguing to consider the possibility of epigenetic effects on aging, but this should only be proposed briefly and in the context of epigenetic inheritance arising from cryptobiosis, as the topic of this review is supposed to be the relationship between the two.

Overall, the review is set up to address the question of how cryptobiosis and anhydrobiosis can impact aging. This point should be emphasized within each of the sections in order to clearly relate each of them to the central question. There are many times where the writing distracts from this central theme.

Minor Comments

Anhydrobiosis literally means “life without water.” Dehydration tolerance, or desiccation tolerance are related. (Lines 93-94)

Line 141 “is not indiscrete” is a particularly confusing phrase.

Line 146 “reactive oxygen species are highly reactive” is redundant.

Is there significance to some arrows in Figure 1 being bold and others not?

How is lifespan scored in the species for which it is reported? Simply activity? Response to stimulus? Metabolic activity? Is there a way to ensure that tardigrades are truly dead and not in a cryptobiotic or quiescent state?

It may be helpful to discuss healthspan in the context of “active lifespan.” (line 306)

Line 341,343 – The use of bisexual can be confusing. I recommend saying dioecious.

Citations should be included for the claims in lines 400-402.

Reviewer 2 Report

I found this review article interesting but wordy and difficult to follow in many parts. Therefore, I recommend publication pending a major revision of the English language. My indication of major revision reflects these points. 

Round 2

Reviewer 1 Report

The revised manuscript is greatly improved. I only have minor suggestions for the authors.

- "Anhydrobiosis effect on aging" is a phrase used throughout the manuscript. I suggest rephrasing this as "the effect of anhydrobiosis on aging."

- This phrase is also used in the title. A slight modification of the title could improve readability: "Applicable life-history and molecular traits for studying the effects of anhydrobiosis on aging in tardigrades"

Reviewer 2 Report

The revised version of the ms is fine. After my last review I noticed that the dr. Guidetti, who runs another tardigrades database, report 1400 species in total. It would be wise to cite this datum too.
